# Semi-supervised Camouflaged Object Detection from Noisy Data

Yuanbin Fu
Tianjin University
Tianjin, China
yuanbinfu@tju.edu.cn

Jie Ying
Inspur Smart City
Technology Co, Ltd
Jinan, China
yingjie@inspur.com

Houlei Lv
Inspur Smart City
Technology Co, Ltd
Jinan, China
lvhoulei@inspur.com

Xiaojie Guo*
Tianjin University
Tianjin, China
xj.max.guo@gmail.com

## Abstract

Most of previous camouflaged object detection methods heavily lean upon large-scale manually-labeled training samples, which are notoriously difficult to obtain. Even worse, the reliability of labels is compromised by the inherent challenges in accurately annotating concealed targets that exhibit high similarities with their surroundings. To overcome these shortcomings, this paper develops the first semi-supervised camouflaged object detection framework, which requires merely a small amount of samples even having noisy/incorrect annotations. Specifically, on the one hand, we introduce an innovative pixel-level loss re-weighting technique to reduce possible negative impacts from imperfect labels, through a window-based voting strategy. On the other hand, we take advantages of ensemble learning to explore robust features against noises/outliers, thereby generating relatively reliable pseudo labels for unlabelled images. Extensive experimental results on four benchmark datasets have been conducted.

## CCS Concepts

• **Computing methodologies → Image manipulation**.

## Keywords

Semi-supervised, Camouflaged object detection, Pseudo labels

**ACM Reference Format:**
Yuanbin Fu, Jie Ying, Houlei Lv, and Xiaojie Guo. 2024. Semi-supervised Camouflaged Object Detection from Noisy Data. In *Proceedings of the 32nd ACM International Conference on Multimedia (MM '24), October 28-November 1, 2024, Melbourne, VIC, Australia.* ACM, New York, NY, USA, 10 pages. https://doi.org/10.1145/3664647.3680645

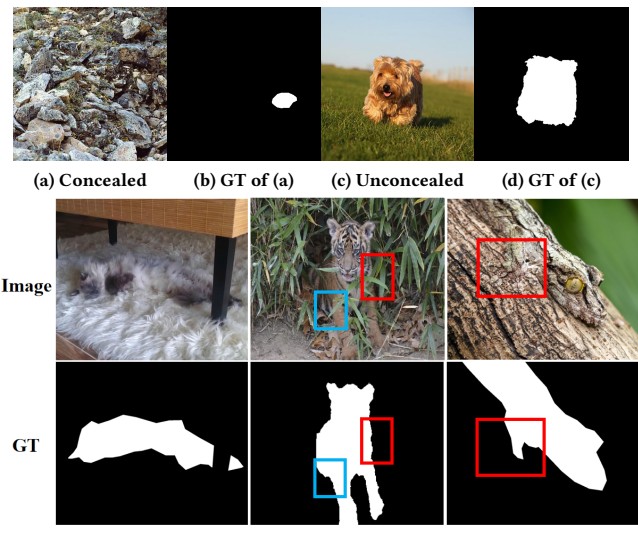

**(a) Concealed** **(b) GT of (a)** **(c) Unconcealed** **(d) GT of (c)**

Image

GT

Figure 1: The top row shows a visual comparison between concealed and unconcealed objects. The concealed object in (a) is more difficult to label than the unconcealed object in (c). The rest two rows provide three cases with noisy annotations: The left-most column displays a case with extremely complex boundary. The middle shows that the boundary of the tiger is corrupted by messy leafs as indicated by red and blue boxes, which does not match with the ground truth, while the right-most one suffers from incorrect annotation.

## 1 Introduction

Camouflaged object detection (COD) aims to seek objects that are cunningly hidden into their ambient regions. It is capable of assisting numerous applications such as search-and-rescue [14], healthcare [16, 17], and recreational art [9], *etc.* In previous decades, traditional methods utilize hand-engineered priors, *e.g.*, orientation gradients [50], scales [18], and/or colors [54], to discover camouflaged objects, the performance of which, however, is barely satisfactory in real scenarios.

Recently, the development in deep learning techniques has made remarkable progress for COD [6, 21, 27, 30, 38, 44, 52, 58, 65], owing to the proficiency in directly learning the mapping from input images to annotated ground truths. But, precisely annotating concealed targets in natural images, especially those with camouflaged colors, ambiguous boundaries, and/or disrupted occlusions, remains challenging. As demonstrated by [15, 25], manually annotating concealed objects is far more arduous than doing normal/unconcealed ones, the cost of which is about one hour per image. For instance, the appearance of the target in Fig. 1 (a) is very similar to its surroundings, requiring tremendous efforts to discover it.

To relieve the pressure from data, the semi-supervised learning is an option, in which the amount of the labelled training set can be smaller than that of the unlabelled set. Over the past decade, numerous semi-supervised learning strategies have been developed [1, 2, 5, 28, 29, 36, 39]. But , their common assumption is that the

*Corresponding author.

manual annotations in the labelled training set are clean and correct, without attention to the noisy/outlying annotations existing in the labelled set. For the COD task, unfortunately, it is hard to avoid noisy/incorrect labels, due to the difficulty in accurately annotating camouflaged objects as aforementioned. Please see Fig. 1 for evidence. Besides, under the semi-supervised setting, the scarcity of labelled samples inevitably increases the risk of over-fitting those detrimental noises/outliers, severely impeding the ability of deep semi-supervised COD models. Thus, a question naturally arises: *how to explore discriminative and robust knowledge from noisy manual labels in a small amount of data, so as to provide reliable guidance, e.g., pseudo labels, for supervising other unlabelled images?*

In this work, we devise a semi-supervised COD framework for mitigating the heavy annotation burden. It accomplishes the semi-supervised learning amidst noisy manual labels by embracing two pivotal techniques including loss re-weighting, and ensemble learning. To be more specific, on the one hand, a novel loss re-weighting strategy is proposed to adjust the penalty strength with respect to each pixel. The principle behind is that, for two neighbor pixels, both of them highly likely belong to camouflaged objects, or neither of them does. If the category of a pixel deviates from that of its nearby pixels, the annotation for this pixel is suspected to be incorrect/noisy. It is imperative to safeguard against undue deviations in the optimization direction, which may arise from the annotations inconsistent with their neighbors. To this end, we propose to determine the loss weights specific to each pixel, according to the neighbor information within a window. On the other hand, we seek to integrate potentially weak-yet-valuable knowledge from different sources, *e.g.*, different models or different training moments. By doing so, the interference of noisy/incorrect annotations inherent in the training data can be alleviated, drawing upon the principles of ensemble learning. Despite the knowledge learned either at a particular training moment, or by a single model, is easily disturbed by noisy/outlier samples, it remains eligible to serve as a valuable source for ensemble. Based on the above, we utilize the information from both different training moments and different models to generate pseudo labels. Notably, our model solely necessitates the backbone pre-trained in a *unsupervised* manner, obviating the requirement for any labeled data from large-scale datasets, *e.g.*, ImageNet [11]. Our project page is available at: https://github.com/ForawardStar/COD, and our contributions can be summarized as follows:

- We develop a semi-supervised learning scheme amidst noisy manual labels, to liberate the heavy burden of annotation. To the best of our knowledge, this work is the first attempt of semi-supervised COD.
- We propose a novel window-based voting strategy to adjust loss weights to each pixel, and an ensemble learning algorithm to integrate the knowledge from different training moments and different networks.
- We customize the network architecture, to advance the integration and interaction across different scales, emulating the behavior of human vision system when viewing complex scenes.
- We conduct extensive experiments to verify the effectiveness of our proposed method on benchmark datasets. The

results indicate that our method can achieve competitive performance compared to fully-supervised COD models.

## 2 Related Work

*Camouflaged Object Detection.* Early attempts on COD typically utilize hand-crafted features to recognize camouflaged objects, the performance of which, however, is limited. Recently, benefiting from the availability of manually labelled datasets, such as, CHAMELEON [55], NC4K [45], CAMO [33], and COD10K [14, 15], the field of camouflaged object detection has been significantly propelled forward by deep learning techniques. As a representative, SINet [14] incorporates a search phase responsible for finding the locations of concealed objects, and an identification phase for precisely segmenting detection targets. FindNet [37] embeds both the boundary and texture information into the learned features, and the boundary and texture enhancement modules are adopted to highlight global and local patterns, respectively. ZoomNet [51] employs a zoom in and out operation to explore mixed-scale semantics, aiming at handling the objects with diverse scales. HitNet [27] refines the low-resolution representations using high-resolution ones, in an iterative feedback fashion. FEDER [21] decomposes the features into multiple frequency bands through learnable wavelets, where the most informative bands are mined to differentiate foreground and background. Similarly, [10, 63, 71] introduce the frequency clue as an extra evidence for recognizing camouflaged objects. MSCAF-Net [41] uses an enhanced receptive field module to refine the features learned at each layer, and a cross-scale feature fusion module to enrich the diversity of extracted features. FSPNet [30] makes use of a non-local token enhancement module that interacts neighboring tokens and explores graph-based high-order relations within tokens. Some works [35, 66, 70] discover difficult-to-detect samples to facilitate the identification of the camouflaged objects that are tough to be precisely found, based on uncertainty estimation. Several other works also [34, 46, 69, 73] leverage the benefits of multi-task learning that introduces auxiliary tasks, such as image classification and edge detection.

Despite exhibiting promising performance, previous deep learning models heavily rely on extensive manually annotated training data, which poses significant challenges in terms of collection. Recently, He *et al.* [25] customized a weakly supervised COD framework, but there is still a large gap in the detection accuracy compared to fully-supervised methods. For the sake of reducing the heavy annotation burden, a semi-supervised COD framework is highly desired, which requires only a limited number of labelled samples. Hence, in this paper, we develop the first semi-supervised COD framework that achieves competitive accruacy compared to previous fully-supervised deep models.

*Semi-Supervised Learning.* Semi-supervised learning serves as the cornerstone of numerous computer vision and multimedia tasks, the goal of which is to learn a deep model using a small amount of labelled data, together with unlabelled data. Generally, semi-supervised learning methods can be roughly divided into two categories: entropy minimization, and consistency regularization.

The approaches based on entropy minimization concern themselves with anticipating unlabelled data exhibiting low entropy. As pointed out by [19], the predictions made for unlabelled samples

should be situated distant from the decision boundary, underlining the imperative of minimizing entropy in unlabelled data. VAT [49] defines a novel virtual adversarial loss as the measurement of local smoothness around every input data points against local perturbations. Wang *et al.* [61] separated certain and uncertain pixels based on the entropy of their predictions, It redirects each unreliable pixel to a category-specific queue populated with negative samples, while striving to train the model using all potential pixels. St++ [67] implements selective re-training by prioritizing reliable unlabeled images, prioritizing those that exhibit holistic prediction-level stability. Dense teacher [72] introduces an innovative region selection technique that effectively highlights crucial information while efficiently suppressing noises associated with dense labels.

Besides entropy minimization, consistency regularization is also effective in semi-supervised learning. As data augmentation demonstrates evident effectiveness, some works center on ensuring the predictions of two augmented/perturbed inputs stay consistent through regularization. For example, CutMix [68] cuts and pastes images patches between two different images, with the ground truth labels mixed proportionally to the area of the patches. UDA [64] delves into the role of noise injection in consistency training and observe that advanced data augmentation techniques, particularly those that excel in supervised learning, hold promise in this context. ReMixMatch [3] employs a distribution alignment strategy aimed at narrowing the gap between the marginal distribution of predictions on unlabeled data and that of ground-truth labels. FixMatch [56] begins by generating pseudo-labels from the model's predictions on weakly augmented unlabeled images. These labels are kept only if the model predicts with high confidence. Subsequently, the model is trained to predict these labels on strongly augmented versions of the same images. Nonetheless, the preset threshold employed in FixMatch relegates certain examples of low confidence to obscurity, preventing them from misleading the learning procedure.

Though achieving promising performance, the above mentioned semi-supervised learning methods usually assume that the labelled data is correct and reliable, without the consideration of noisy annotations. However, there exist severe noisy annotations in the COD datasets, posing a considerable obstacle to learn reliable knowledge from the labelled set. It is infeasible to directly apply existing semi-supervised methods to COD. Hence, in this paper, we develop a novel semi-supervised COD method.

## 3 Methodology

We denote the labelled training data by $\mathbf{D}_l := \{(X_n, Y_n), n := 1, 2, ..., N\}$, where $N$ is the total number of labelled training samples, $X_n$ is the natural images used for input, and $Y_n$ is the corresponding ground truth. In the semi-supervised setting, the number of samples in the labelled training set $\mathbf{D}_l$ is smaller than that in the unlabelled set $\mathbf{D}_u := \{X_m, m := 1, 2, ..., M\}$, *i.e.*, $N < M$. This work aims to devise a semi-supervised camouflaged object detection framework, by considering the adverse influence of noisy manual annotations exist in the labelled training set $\mathbf{D}_l$. To achieve the goal, we propose to 1) re-weight the losses specific/particular to each pixel, based on the neighbor information, and 2) integrate the knowledge across different training moments and networks.

---

**Algorithm 1** Ensemble Learning for Semi-Supervised COD

---

**Input**: Training dataset $\mathbf{D}_l$ and $\mathbf{D}_u$; the total number of labelled samples $N$, and unlabelled samples $M$; the total number of training epochs $EP$; two networks $G^1$ and $G^2$ trained by back-propagation; two momentum networks $G_{mo}^1$ and $G_{mo}^2$; data augmentation function DataAug($\cdot$)

**Output**: The network with the minimum average training loss

**Initialize**: $ep := 0$; Randomly initialize $G^1$ and $G^2$

1: **for** $ep \leftarrow 1$ to $EP$ **do**
2:     Compute $\alpha_{ep}$ by $\alpha_{ep} := ep/EP$;
3:     $num := 0$;
4:     $\mathbf{S}_l := \{n|1, 2, ..., N\}$, $\mathbf{S}_u := \{m|1, 2, ..., M\}$;
5:     **for** $num \leftarrow 1$ to $N$ **do**
6:         Uniformly sample the indexes $n$ and $m$ from $\mathbf{S}_l$ and $\mathbf{S}_u$, respectively;
7:         Read data $X_m$ from $\mathbf{D}_u$;
8:         $O_m^1 := G_{mo}^1(X_m)$, $O_m^2 := G_{mo}^2(X_m)$;
9:         Compute $\widetilde{Y}_m$ by Eq. (4);
10:        Read data $X_n, Y_n$ from $\mathbf{D}_l$;
11:        $X_n^1 := \text{DataAug}(X_n)$, $X_n^2 := \text{DataAug}(X_n)$;
12:        $V_n^1 := G^1(X_n^1), C_m^1 := G^1(X_m)$;
13:        Compute $\mathcal{L}_l^1$ by Eq. (8) and $\mathcal{L}_u^1$ by Eq. (10);
14:        Update $G^1$ through back-propagation;
15:        $V_n^2 := G^2(X_n^2), C_m^2 := G^2(X_m)$;
16:        Compute $\mathcal{L}_l^2$ by Eq. (9) and $\mathcal{L}_u^2$ by Eq. (11);
17:        Update $G^2$ through back-propagation;
18:        Remove $n$ and $m$ from $\mathbf{S}_l$ and $\mathbf{S}_u$, respectively;
19:     **end for**
20:     **if** The average training loss of $G^1$ decreases **then**
21:         Update $G_{mo}^1$ by Eq. (3);
22:     **else**
23:         Keep the parameters of $G_{mo}^1$ unchanged;
24:     **end if**
25:     **if** The average training loss of $G^2$ decreases **then**
26:         Update $G_{mo}^2$ by Eq. (3);
27:     **else**
28:         Keep the parameters of $G_{mo}^2$ unchanged;
29:     **end if**
30: **end for**

---

### 3.1 Loss Re-weighting Strategy

As discussed above, the annotated learning target $Y_n$ may contain noisy or wrong annotations, which are highly likely to mislead the optimization direction during training. To reduce the disturbance of noisy labels in $\mathbf{D}_l$, we design a window-based voting strategy, to adjust the penalty strength for each pixel, according to the neighbor information within a window. In the context of dense pixel-wise classification tasks, *e.g.*, COD or semantic segmentation, a ubiquitous observation is that pixels in close proximity usually shows similar colors, textures, or other visual characteristics, tending to belong to the same category or object. The reason lies in that pixels in natural images do not typically exist in isolation, but instead are correlated with their neighboring pixels. Such inter-connectivity often stems from the continuity of objects, and the consistency of visual effects such as appearances and depths. Based on this, we

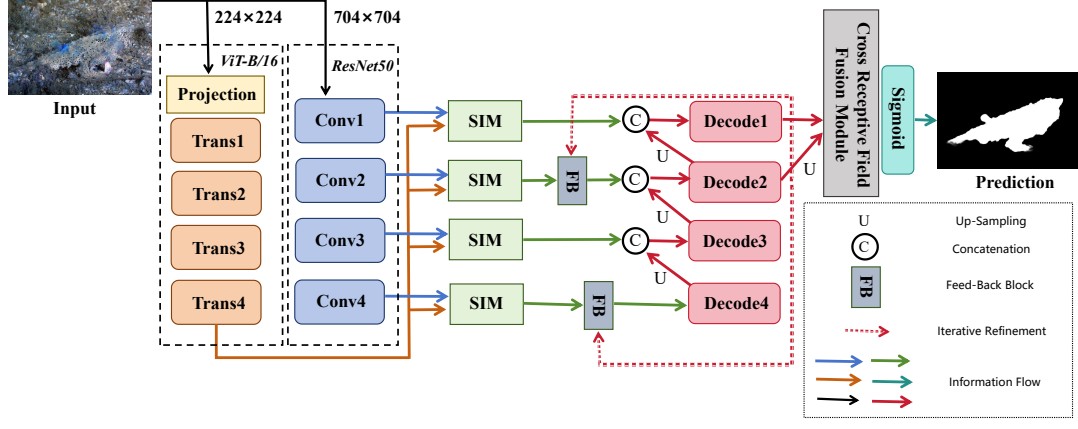

**(a) Overall Architecture**

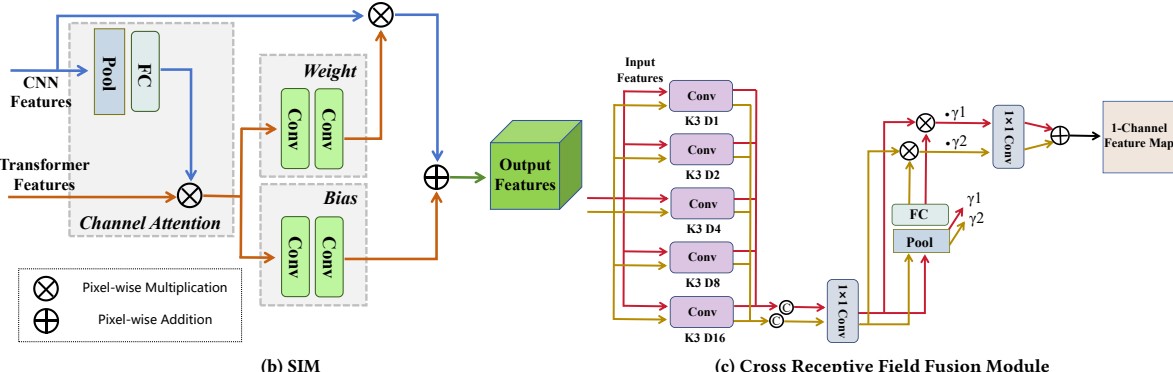

**(b) SIM**   **(c) Cross Receptive Field Fusion Module**

**Figure 2: The illustration of our network architecture. "K" and "D" below each convolution block in (c), indicate the kernel size and dilatation rate, respectively.**

argue that if the classification of a pixel has a significant deviation from its neighbors, the reliability of its annotation is low.

Therefore, to mitigate the misleading effects of these unreliable pixels on optimization direction, we propose a window-based voting strategy that determines the loss weights of a pixel based on its deviation from the voting results of adjacent pixels. The loss weight for the $i$-th pixel in a label $Y$ can be calculated as:

$$\omega(Y^{(i)}) := 1 - |Y^{(i)} - \text{Vot}(Y^{(i)})|, \tag{1}$$

where $|\cdot|$ is the absolute operation, and the values in $Y$ are bounded by 0 as the lower bound and 1 as the upper bound. The voting function $\text{Vot}(\cdot)$ is to enable pixels within a designated window to cast their votes. It can be formulated as:

$$\text{Vot}(Y^{(i)}) := \frac{1}{|\mathcal{W}(i)|} \sum_{j \in \mathcal{W}(i), j \neq i} Y^{(j)}, \tag{2}$$

where $\mathcal{W}(i)$ denotes a squared window centered at the $i$-th pixel, and $|\mathcal{W}(i)|$ is the total amount of pixels within the window $\mathcal{W}(i)$.

## 3.2 Information Integration

For the sufficient capacity in exploring robust features in the noisy labelled set $\mathbf{D}_l$, we propose a novel ensemble learning algorithm as detailed in **Algorithm** 1, which integrates the knowledge learned from different training moments and different models. The proposed ensemble learning algorithm helps to provide reliable pseudo labels for the unlabelled set $\mathbf{D}_u$. In testing, only one network with the minimum average training loss throughout the entire training process, is executed. As evidenced by the literature on model ensemble [8, 26, 43], integrating predictions from diverse sources is reasonable, due to that it aids in averting over-fitting to noises/outliers.

To integrate the knowledge from different training moments, we incorporate the momentum network whose parameters are updated by accumulating the parameters of the back-propagation network trained after different epochs. But, simply aggregating the knowledge learned at all epochs, leads to sub-optimal detection accuracy as some unwanted epochs with poor performance are also involved for integration. Thus, to mitigate the disruption caused by undesirable training moments, we propose to discard those unwanted epochs that yield unsatisfactory results, through assessing whether the losses decrease in comparison to the preceding epochs. In detail,

**Table 1: Quantitative comparison with state-of-the-arts on four datasets, *i.e.*, CAMO, CHAMELEON, COD10K, and NC4K. The best results are highlighted in Bold, while the second best results are marked in *Italic*. "Ours-10%", "Ours-25%", and "Ours-40%" mean that 10%, 25%, and 40% samples of the total training data are labelled, respectively, while other samples are unlabelled. "Ours-all" indicates that all training samples in COD datasets are labelled, yet the backbone is initialized using the parameters pre-trained in a fully-supervised manner. The results of other competitors are directly cited from their original papers.**

| Method | CAMO | | | | CHAMELEON | | | | COD10K | | | | NC4K | | | |
|---|---|---|---|---|---|---|---|---|---|---|---|---|---|---|---|---|
| | MAE↓ | $S_m$↑ | $F_\beta^w$↑ | $E_m$↑ | MAE↓ | $S_m$↑ | $F_\beta^w$↑ | $E_m$↑ | MAE↓ | $S_m$↑ | $F_\beta^w$↑ | $E_m$↑ | MAE↓ | $S_m$↑ | $F_\beta^w$↑ | $E_m$↑ |
| Fully-Supervised | | | | | | | | | | | | | | | | |
| ANet [34] | .126 | .682 | .484 | .722 | - | - | - | - | - | - | - | - | - | - | - | - |
| SINet [15] | .092 | .745 | .644 | .829 | .034 | .872 | .806 | .946 | .043 | .776 | .631 | .874 | .058 | .808 | .723 | .883 |
| SINetV2 [14] | .070 | .820 | .743 | .895 | - | - | - | - | .037 | .815 | .680 | .906 | .048 | .847 | .770 | .914 |
| TINet [75] | .087 | .781 | .678 | .848 | - | - | - | - | .042 | .793 | .635 | .878 | .055 | .829 | .734 | .890 |
| SLSR [45] | .080 | .787 | .696 | .854 | .030 | .890 | .822 | .948 | .037 | .804 | .673 | .892 | .048 | .840 | .766 | .907 |
| MGL [69] | .088 | .775 | .673 | .842 | .031 | .893 | .812 | .941 | .035 | .814 | .666 | .890 | .053 | .833 | .739 | .893 |
| PFNet [48] | .085 | .782 | .695 | .855 | .033 | .882 | .810 | .945 | .040 | .800 | .660 | .890 | .053 | .829 | .745 | .898 |
| UJSC [35] | .073 | .800 | .728 | .873 | .030 | .891 | .833 | .955 | .035 | .809 | .684 | .891 | .047 | .842 | .771 | .907 |
| C2FNet [57] | .080 | .796 | .719 | .864 | .032 | .888 | .828 | .946 | .036 | .813 | .686 | .900 | .049 | .838 | .762 | .904 |
| UGTR [66] | .086 | .784 | .684 | .851 | .031 | .888 | .794 | .940 | .036 | .817 | .666 | .890 | .052 | .839 | .746 | .899 |
| ZoomNet [51] | .066 | .820 | .752 | .892 | .023 | .902 | .845 | *.958* | .029 | .838 | .729 | .911 | .043 | .853 | .784 | .912 |
| FEDER [21] | .066 | .836 | - | - | .026 | .903 | - | - | .029 | .844 | - | - | .042 | .862 | - | - |
| FSPNet [30] | **.050** | **.856** | .799 | *.928* | - | - | - | - | .026 | .851 | .735 | .930 | *.035* | *.879* | .816 | *.937* |
| SegMaR [31] | .071 | .815 | .742 | .884 | .025 | .906 | .860 | - | .033 | .833 | .724 | .906 | .046 | .841 | .781 | .907 |
| Frequency [71] | .062 | .844 | .778 | - | .027 | .898 | .837 | - | .030 | .837 | .731 | - | - | - | - | - |
| DaCOD [59] | .051 | .855 | .796 | - | - | - | - | - | .028 | .840 | .729 | - | .035 | .874 | .814 | - |
| PENet [38] | .063 | .828 | .771 | - | .024 | .902 | .851 | - | .031 | .831 | .723 | - | .042 | .855 | .795 | - |
| BGNet [58] | .073 | .812 | .749 | .882 | - | - | - | - | .033 | .831 | .722 | .911 | .044 | .851 | .788 | .916 |
| FindNet [73] | .079 | .796 | .717 | .867 | .027 | .895 | .841 | - | .034 | .818 | .699 | .901 | .048 | .841 | .771 | .907 |
| HitNet [27] | .056 | .844 | *.806* | .910 | *.018* | *.922* | *.903* | - | *.023* | *.869* | *.804* | *.938* | .039 | .870 | *.825* | .929 |
| Ours-all | *.051* | *.854* | **.817** | **.932** | **.016** | **.928** | **.907** | **.975** | **.020** | **.874** | **.812** | **.940** | **.033** | **.887** | **.840** | **.939** |
| Weakly-Supervised | | | | | | | | | | | | | | | | |
| Scribble [25] | .092 | .735 | .641 | - | .046 | .818 | .744 | - | .049 | .733 | .576 | - | - | - | - | - |
| Semi-Supervised | | | | | | | | | | | | | | | | |
| Ours-10% | .077 | .789 | .732 | .859 | .036 | .850 | .773 | .928 | .033 | .819 | .725 | .891 | .046 | .838 | .787 | .903 |
| Ours-25% | .074 | .796 | .737 | .866 | .034 | .858 | .780 | .934 | .032 | .824 | .728 | .896 | .044 | .844 | .795 | .911 |
| Ours-40% | .071 | .805 | .746 | .871 | .032 | .865 | .787 | .940 | .029 | .830 | .735 | .905 | .041 | .851 | .802 | .918 |

as illustrated in **Algorithm** 1, if the average losses at the current epoch decrease compared to the last epoch, the parameters of the momentum network can be updated through:

$$\mathbf{P}_{mo}^{ep} := (\mathbf{P}_{bp}^{ep} + \mathbf{P}_{mo}^{ep-1})/2, \tag{3}$$

where $\mathbf{P}_{bp}^{ep}$ and $\mathbf{P}_{mo}^{ep}$ indicate the parameters of the back-propagation network and momentum network at $ep$-th epoch, respectively; Otherwise, the parameters remain unchanged, say $\mathbf{P}_{mo}^{ep} := \mathbf{P}_{mo}^{ep-1}$.

As for combining the features from different models, we incorporate two networks, *i.e.*, $G^1(\cdot)$ and $G^2(\cdot)$, with identical architectures as detailed in Section 3.3. Both of $G^1(\cdot)$ and $G^2(\cdot)$ are trained via back-propagation, along with two momentum networks $G_{mo}^1(\cdot)$ and $G_{mo}^2(\cdot)$ corresponding to $G^1(\cdot)$ and $G^2(\cdot)$, respectively. Note that $G^1(\cdot)$ and $G^2(\cdot)$ are initialized with random distinctive parameters, and receive different augmented inputs, which contributes

to capture diverse information for ensemble. To generate pseudo labels for samples in the unlabelled set $\mathbf{D}_u$, the predictions by two momentum networks will be fused according to their pixel-level uncertainties. The process of producing the pseudo label $\widetilde{Y}_m$ for the $m$-th unlabelled sample, can be described as:

$$\widetilde{Y}_m := U_m^1 \circ O_m^1 + U_m^2 \circ O_m^2, \tag{4}$$

in which

$$U_m^1 := \frac{|O_m^1 - 0.5|}{|O_m^1 - 0.5| + |O_m^2 - 0.5|},$$
$$U_m^2 := \frac{|O_m^2 - 0.5|}{|O_m^1 - 0.5| + |O_m^2 - 0.5|}, \tag{5}$$

where $\circ$ denotes the Hadamard product, $O_m^1 := G_{mo}^1(X_m)$ and $O_m^2 := G_{mo}^2(X_m)$, $U_m^1$ and $U_m^2$ are the uncertainty scores of the predictions $O_m^1$ and $O_m^2$, respectively. If the predictions by a network

hover nearer (further) to 0.5 in comparison to another, it signifies a greater (lesser) degree of uncertainty in its predictions. As a result, its influence in the fusion process should be diminished (enhanced). It is worth to mention that the images in the labelled set $\mathbf{D}_l$ will not be processed by our momentum networks.

## 3.3 Network Architecture

For the COD task, it is critical to make use of the information from different scales, imitating the human behaviors when perceiving perplexing scenes. Otherwise, the detection results would be considerably poor when handling the samples with low reliability in annotations. As demonstrated by [30, 38, 51], to pinpoint camouflaged objects in a scene, humans often reference and compare changes in shape or appearance across different scales, which supports the design of numerous deep COD models. To effectively achieve the goal of capturing the mixed-scale semantics, we employ a U-shaped network architecture as shown in Fig. 2 (a). We advocate that small-scale features can be extracted by a convolution neural network (CNN) due to its local nature [24, 40, 60, 62, 74], and large-scale features can be explored by a vision transformer (ViT) thanks to its ability to model global relationships among patch tokens [4, 12, 20, 32, 42, 53].

Hence, we take advantages of ViT-B/16 [12] and ResNet50 [24] as our backbone to explore mixed-scale information, the input sizes of which are $224 \times 224$ and $704 \times 704$, respectively. We emphasize that ViT-B/16 and ResNet50 are respectively pre-trained through leveraging MAE [22] and MoCo [23] in a unsupervised fashion, without relying on any labels in ImageNet [11]. The deepest block of ViT-B/16 learns features with larger scales, which are then leveraged to individually engage and interact with the smaller-scale features captured by four convolution blocks of ResNet50. For the sake of ensuring sufficient interaction between the features learned by CNN and ViT, as shown in Fig. 2 (b), a scale interaction module (SIM) is designed. The channel attention scores predicted according to the smaller-scale features extracted by CNN (brown arrows in Fig. 2 (b)), are utilized to steer the larger-scale features acquired by ViT (blue arrows in Fig. 2 (b)). These steered features are then employed to predict the scaling weights for pixel-wise multiplication with smaller-scale features, as well as the biases for addition. Inspired by [27], two feed-back blocks are used for iteratively refining the features of the second and forth blocks, which progressively enlarge the receptive fields without increasing the parameter amount. Subsequently, after being interacted using our SIM and refined by feed-back blocks, the decoder takes these features as input to produce multi-scale (four scales in total) representations. The features generated by the 1-st block together with the features refined by feed-back blocks will be fused using our proposed cross receptive field fusion module (CRFM). As shown in Fig. 2 (c), we draw support from dilated convolutions with exponentially increased dilation rates, to excavate the features of different receptive fields. After being processed by a channel attention operation for enhancing useful information, a $1 \times 1$ convolution for outputting a one-channel feature map in Fig. 2 (c), and a Sigmoid activation in Fig. 2 (a), final results can be obtained. The red and orange line in Fig. 2 (c), represent the two groups of features that are inputted into our CRFM, respectively.

## 3.4 Learning Objective

The total learning objective consists of the weighted Binary Cross Entropy (BCE) and Intersection over Union (IOU) loss, which can be formulated as:

$$\mathcal{L}_{bce}(R, Y) := -\sum_{i}^{|Y|} \omega(Y^{(i)}) \cdot [Y^{(i)} \text{Log}(R^{(i)}) + (1 - Y^{(i)}) \text{Log}(1 - R^{(i)})],$$
(6)

$$\mathcal{L}_{iou}(R, Y) := \sum_{i}^{|Y|} 1 - \frac{R \circ Y}{R + Y},$$
(7)

where $R$ and $Y$ are the detection result and ground truth, respectively. $|Y|$ is the total number of pixels in $Y$. For samples in the labelled set $\mathbf{D}_l$, Eq. 8 and Eq. 9 are the losses of $G^1(\cdot)$ and $G^2(\cdot)$, respectively, which are formulated as:

$$\mathcal{L}_l^1 := \sum_{n:=1}^{N} [\mathcal{L}_{bce}(V_n^1, Y_n) + \mathcal{L}_{iou}(V_n^1, Y_n)],$$
(8)

$$\mathcal{L}_l^2 := \sum_{n:=1}^{N} [\mathcal{L}_{bce}(V_n^2, Y_n) + \mathcal{L}_{iou}(V_n^2, Y_n)],$$
(9)

where $V_n^1 := G^1(X_n)$, and $V_n^2 := G^2(X_n)$. For the unlabelled samples in $\mathbf{D}_u$, due to that the performance of deep models is poor at the early training stage, the pseudo labels generated through Eq. 4 are not so reliable. To avoid being misled by low-quality pseudo labels, we gradually increase the weight of the losses as training proceeds. Therefore, the losses for unlabelled images are formulated as:

$$\mathcal{L}_u^1 := \sum_{m:=1}^{M} \alpha_{ep} \cdot [\mathcal{L}_{bce}(C_m^1, \widetilde{Y}_m) + \mathcal{L}_{iou}(C_m^1, \widetilde{Y}_m)],$$
(10)

$$\mathcal{L}_u^2 := \sum_{m:=1}^{M} \alpha_{ep} \cdot [\mathcal{L}_{bce}(C_m^2, \widetilde{Y}_m) + \mathcal{L}_{iou}(C_m^2, \widetilde{Y}_m)],$$
(11)

where $C_m^1 := G^1(X_m)$, and $C_m^2 := G^2(X_m)$, and $\alpha_{ep}$ is the weight that controls how much we are going to trust pseudo labels at $ep$-th epoch. Denote by $EP$ the total number of training epochs, $\alpha_{ep}$ is calculated by $\alpha_{ep} := ep/EP$.

## 4 Experimental Validation

### 4.1 Implementation Details

We implement our network using PyTorch library on a RTX 3090 GPU. In detail, both of $G^1(\cdot)$ and $G^2(\cdot)$ are randomly initialized and collaboratively trained for 80 epochs using AdamW optimizer. The momentum and weight decay of the optimizer are set to 0.9 and 0.0001, respectively. The gradients of back-propagation are clipped so that they are limited to the range of a closed interval [-0.5 0.5]. By doing so, the risk of gradient explosion can be reduced. The size of the window $\mathcal{W}$ in Eq. 2 is $31 \times 31$. During training, we randomly rotate the image between -180 and 180 degrees, randomly perturb the brightness, contrast, saturation, and hue of images, from 50% to 150% of their original values, and randomly convert input images to gray-scale images in a probability of 0.2, for data augmentation. In this way, two differently augmented images will be produced as the inputs of $G^1(\cdot)$ and $G^2(\cdot)$, respectively.

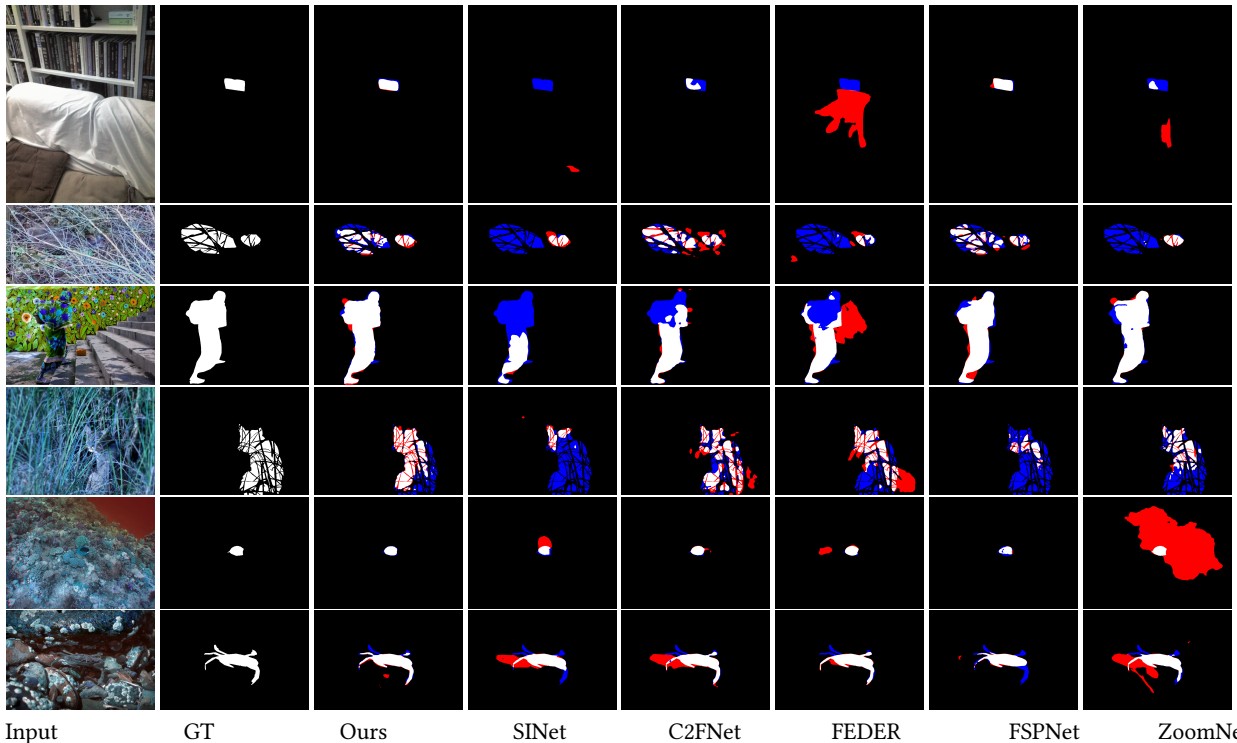

|       | Input | GT | Ours | SINet | C2FNet | FEDER | FSPNet | ZoomNet |

**Figure 3: Visual comparison with other fully-supervised COD methods. The red and blue regions represent false-positive and false-negative predictions, respectively.**

**Table 2: Ablation studies on four datasets. The best results are highlighted in Bold, while the second best results are marked in *Italic*. All the results are trained through selecting 40% from the training dataset as the labelled samples, and the remaining data is unlabelled.**

| Method | CAMO | | | | CHAMELEON | | | | COD10K | | | | NC4K | | | |
|---|---|---|---|---|---|---|---|---|---|---|---|---|---|---|---|---|
| | MAE↓ | $S_m$ ↑ | $F_\beta^w$ ↑ | $E_m$ ↑ | MAE↓ | $S_m$ ↑ | $F_\beta^w$ ↑ | $E_m$ ↑ | MAE↓ | $S_m$ ↑ | $F_\beta^w$ ↑ | $E_m$ ↑ | MAE↓ | $S_m$ ↑ | $F_\beta^w$ ↑ | $E_m$ ↑ |
| w/o $\mathbf{D}_u$ | .078 | .793 | .731 | .836 | .042 | .840 | .734 | .921 | .042 | .796 | .701 | .868 | .053 | .809 | .734 | .886 |
| w/o re-weighting | .075 | .801 | .742 | *.856* | .039 | .846 | .767 | .927 | .034 | .819 | .723 | .889 | .046 | .844 | .781 | .897 |
| w/o fusing models & moments | .079 | .784 | *.744* | .842 | .040 | .839 | .768 | .919 | .036 | .817 | .717 | .873 | .051 | .815 | .739 | .895 |
| w/o fusing models | .074 | .802 | .741 | .853 | .037 | .851 | .776 | *.930* | .032 | *.826* | .728 | .888 | .047 | *.847* | .785 | .890 |
| w/o fusing moments | .076 | .792 | .739 | .852 | .039 | .844 | .774 | .924 | .033 | .822 | .725 | .889 | .045 | .845 | .783 | .903 |
| w/o high-level | *.073* | *.803* | .738 | .855 | *.034* | *.861* | .781 | .929 | .032 | .821 | *.729* | .893 | *.043* | .846 | .793 | .904 |
| w/o SIM | .077 | .798 | .735 | .850 | .036 | .856 | .780 | .929 | *.030* | .823 | .726 | .892 | .047 | .841 | *.795* | .905 |
| w/o CRFM | .075 | .800 | .740 | .854 | .035 | .860 | *.782* | .927 | .031 | .825 | .728 | *.894* | .044 | .844 | .793 | *.906* |
| **Ours** | **.071** | **.805** | **.746** | **.871** | **.032** | **.865** | **.787** | **.940** | **.029** | **.830** | **.735** | **.905** | **.041** | **.851** | **.802** | **.918** |

## 4.2 Datasets & Evaluation Criteria

Four datasets are employed in this paper, which include: CAMO [33] having 2,500 samples with eight categories, CHAMELEON [55] with 76 high-resolution images from the Internet, COD10K [14, 15] containing 10,000 images, and NC4K [45] consisting of 4,121 images. Following previous works [14, 21, 25, 30, 51, 52], 3040 samples in COD10K [14] and 1000 samples from CAMO [33] are used for training, while 76 images from CHAMELEON, 250 images from CAMO, 2,026 images from COD10K, and 4,121 images from NC4K, are used for testing.

As for the evaluation metrics, we evaluate all the COD models in this paper with regard to four metrics: 1) mean absolute error (MAE); 2) S-measure ($S_m$) [7]; 3) Weighted F-score ($F_\beta^w$) [47]; 4) E-measure ($E_m$) [13].

Note that owing to the absence of a thoroughly clean and correct test set, we are necessarily constrained to utilizing noisy labeled test sets for evaluation purposes. This evaluation approach is deemed equitable as we adhere to the utilization of identical evaluation datasets as employed by other competitors.

 Yuanbin Fu, Jie Ying, Houlei Lv, and Xiaojie Guo

**Table 3: Parameter study on COD10K. The best results are highlighted in bold. All the results are trained through selecting 40% from the training dataset as the labelled samples.**

| Window Size | MAE↓ | $S_m$ ↑ | $F_\beta^w$ ↑ | $E_m$ ↑ |
|:---:|:---:|:---:|:---:|:---:|
| $7 \times 7$ | .033 | .821 | .717 | .892 |
| $15 \times 15$ | .031 | .824 | .720 | .897 |
| $31 \times 31$ | **.029** | **.830** | **.735** | **.905** |
| $63 \times 63$ | .030 | .826 | .724 | .895 |

## 4.3 Comparison with State-of-the-arts

We randomly select 10%, 25%, and 40% of samples from the training datasets as described in Section 4.2, to constitute the labelled training set $\mathbf{D}_l$, while the remaining samples are designated as the unlabelled set $\mathbf{D}_u$. It can be seen from Tab. 1 that our proposed method can achieve competitive accuracy compared with other fully-supervised COD methods. For example, when only 40% samples of the total training data are labelled, our method demonstrates a noteworthy performance, achieving the weighted F-score $F_\beta^w$ of 0.735 on COD10K. Compared with the work for weakly-supervised COD [25], we have a evident advantage in accuracy. We also show the visual results in Fig. 3. Obviously, our method exhibits a remarkable ability to accurately segment concealed objects while minimizing redundant aesthetic predictions in regions exterior to the objects. Please notice that, if all the data is fed into the network, our method becomes fully-supervised. Under this setting, it attains state-of-the-art accuracy over other fully-supervised approaches, as demonstrated by the results indicated as "Ours-all" in Tab. 1.

## 4.4 Ablation Study

To verify the effectiveness of the proposed loss re-weighting and ensemble learning, we report the results of five alternatives: 1) **w/o $\mathbf{D}_u$**. Training a deep COD model using only the samples in the labelled training set $\mathbf{D}_l$, without using any sample in the unlabelled $\mathbf{D}_u$; 2) **w/o re-weighting**. Abandoning the proposed window voting strategy for loss re-weighting, that is, the loss weights calculated in Eq. 1 is abandoned; 3) **w/o fusing models & moments**. Generating the pseudo label $\widetilde{Y}_m$ neither based on knowledge from different training moments nor different models. The knowledge learned by a single model trained after the last epoch, are used to produce $\widetilde{Y}_m$; 4) **w/o fusing models**. Generating the pseudo label $\widetilde{Y}_m$ only based on the knowledge across different training moments. In this setting, only a back-propagation network and its corresponding momentum network are maintained; 5) **w/o fusing moments**. Generating $\widetilde{Y}_m$ only based on the knowledge across different models without maintaining momentum networks, where the knowledge learned from the last epoch, are used to obtain $O_m^1$ and $O_m^2$.

It can be seen from Tab. 2 that, the accuracy of training without using the data in the unlabelled set $\mathbf{D}_u$ is the worst one among all the alternatives, due to the insufficient samples that the network can access during training. Though slightly better than training without using unlabelled data in $\mathbf{D}_u$, the accuracy of generating pseudo labels using merely a single network at a particular moment, still lags behind integrating the knowledge from either different moments or

different models. By amalgamating knowledge from both of these two aspects, the best performance can be obtained, demonstrating the necessity of integrating the knowledge from different sources. Moreover, abandoning the proposed loss re-weighting technique leads to a significant decrease in detection accuracy. This decrement is attributed to the fact that the biased/noisy information present in pixels whose annotations are discordant with their neighbors, is highly prone to misleading the network.

We also discuss the design of our network structure, through compare the performance with: 1) **w/o high-level**. Replacing the features of the deepest transformer block with that of multiple transformer blocks from bottom to top, to correspondingly interact with the features of CNN. The features generated by the 1-st transformer block are taken to interact with the features generated by the 1-st CNN blocks, and the features generated by the 2-nd, 3-rd, and 4-th transformer blocks are taken to interact with the features by the 2-rd, 3-th, and 4-th CNN blocks, respectively; 2) **w/o SIM**. Replacing the designed SIM with a $1 \times 1$ convolution; 3) **w/o CRFM**. Replacing the proposed cross receptive field fusion module with a $1 \times 1$ convolution. As shown in Tab. 2, these variants degrade the detection accuracy, which performs worse than our method.

## 4.5 Parameter Study

This section reports the results of tuning the window size of $\mathcal{W}$ in Eq. 2, to discuss the influence of this hyper-parameter. We can infer from Tab. 3 that, if the window size is small, e.g., $7 \times 7$, the performance is poor, due to that a small window size approximates the absence of a window in its functionality. When gradually increasing the window size to $31 \times 31$, the performance increases accordingly. But, when the window size is larger than $31 \times 31$, the performance will not be further improved. The explanation for this observation is that, some irrelevant and detrimental pixels are involved for voting, if the window size is too large.

## 5 Conclusion

We customized a semi-supervised COD framework to relieve the heavy annotation burden, which is helpful for the COD task being difficult to annotate. To the best of our knowledge, this work is the first attempt for developing a semi-supervised deep COD model. For the sake of learning relative robust knowledge from a small amount of noisy labelled data, a novel loss re-weighting scheme and an ensemble learning algorithm were proposed. In addition, we also carefully designed the network architecture to advance the integration and interaction across different scales. Extensive experiments have been conducted on several public datasets, to reveal that our method can achieve competitive performance compared to other fully-supervised deep COD models. In the future, we will continue to explore the potential of our semi-supervised pipeline on other dense prediction tasks, e.g., edge detection, salient object detection, and semantic segmentation.

## Acknowledgements

This work was supported by the National Natural Science Foundation of China under Grant nos. 62372251 and 62072327.

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
