# OpenReview forum: "Semi-supervised Camouflaged Object Detection from Noisy Data"
_acmmm.org/ACMMM/2024/Conference — MM2024 Poster_

### Official Review · Reviewer_VQmM · 2024-04-29

**Rating:** 4
**Confidence:** 2

**Summary:**

This paper proposes a semi-supervised Camouflaged Object Detection (COD) framework. Pixel-level loss reweighting technique: Reduce the possible negative impact of imperfect labels through a window-based voting strategy.
Ensemble learning: Use ensemble learning to explore features that are robust to noise/outliers and generate relatively reliable pseudo-labels for unlabeled images. Extensive experiments on benchmark datasets demonstrate the competitiveness of the proposed method compared with fully supervised COD models.

**Strengths:**

1.The Pixel-level loss reweighting method is more clever, but the integrated learning using two network fusion feature designs is too bloated.
2.The paper proposes for the first time the use of semi-supervised learning to solve the problems of noisy labels and difficulty in obtaining labels in Camouflaged Object Detection.
3.The paper makes a lot of visual comparisons to highlight the advantages of the method compared with traditional methods.

**Limitations:**

1.Lack of comparison of network structure parameters. The correlation between the amount of network parameters and performance lacks necessary analysis.

2.Why does the SIM module use the features of CNN to weight the features of Transformer? I am confused about the theory of this design.

3.Only 40% was used to achieve competitive experimental results compared to fully supervised training. Can it surpass the fully supervised results if 50% or more is used but does not reach 100%?

**Suitability:**

2

---

### Official Review · Reviewer_f1WD · 2024-04-30

**Rating:** 4
**Confidence:** 2

**Summary:**

This paper proposes the Semi-Supervised Camouflaged Object Detection (COD) framework for the first time to solve the training challenges caused by noisy manual labels. This paper proposes a novel window-based voting strategy to adjust the loss weight of each pixel, and an ensemble learning algorithm to integrate knowledge from different training moments and different networks. Experimental results show that the method can achieve competitive performance compared to fully supervised COD models.

**Strengths:**

The paper is clearly written, the drawings are standardized, and the motivation and method of the article can be understood.

Pixel-level loss reweighting technique and Ensemble learning are proposed to improve the robustness of pseudo-labels in the semi-supervised learning process.

The paper cited a large number of methods and conducted comparative experiments, and contributed a large number of ablation experiments to demonstrate the effectiveness of the method.

**Limitations:**

The network structure design is relatively complex, and the network uses a combination of resnet and vit. The model has a large number of parameters and high complexity, which introduces unfair comparisons.

Visualize the characteristics of the middle layer of resnet and Vit to explain more clearly the complementarity of different network characteristics. In addition, feature visualization at different scales is also necessary.

Comparisons with general semi-supervised segmentation methods in semi-supervised experimental settings are lacking.

**Suitability:**

2

---

### Official Review · Reviewer_RxcE · 2024-05-24

**Rating:** 4
**Confidence:** 4

**Summary:**

This paper attempts to improve the generalization of Video Action Recognition models when facing novel actions. (Existing approaches have explored this problem, but most of these solutions enhance generalization through the augmentation of visual embeddings with temporal information.) This paper approaches the problem from the perspective of human cognitive processes to improve generalization. It provides two main contributions: (1) proposing Action-conditioned Prompts by simulating human cognitive processes, which offer detailed fine-grained descriptions of videos from multiple perspectives; (2) implementing the Multi-modal Action Knowledge Alignment module to match videos with fine-grained textual descriptions. Experiments demonstrate the superiority of the proposed method and the effectiveness of the two modules.

**Strengths:**

* The motivation is clearly described, and the relationship between the problem to be solved and the methods and experiments is clear.
* Constructing a Hierarchical Attribute Graph to use LLM for generating fine-grained textual descriptions is a promising orthogonal approach.
* Compared to previous methods such as attention pooling, the Multi-modal Action Knowledge approach offers better interpretability and is simple and effective.
* The effectiveness of the proposed approach is validated through comparative experiments and ablation studies.

**Limitations:**

1. Using LLMs to provide fine-grained textual descriptions to improve the retrieval performance of matching problems is not a new concept, e.g. [1] has explored the effectiveness of this approach based on classification problems.

   [1] Visual Classification via Description from Large Language Models

2. Unclear definitions of ``novel actions". For example, In lines 149-154 of the article, it is argued that existing methods perform poorly in "novel actions recognition," and as an example, a video of "making sushi" is used as an example, showing its low CLIP matching score.
However, ``making sushi" could be regarded as a more fine-grained activity of ``cooking", from this aspect such "novel action recognition" could be a problem of ``fine-grained action recognition".  Moreover,  in lines 161-165, "novel actions" are described as "previously unencountered actions," which is considered somewhat overly generalized.

3. Following the points in 2, such a proposed hierarchical structure for annotating actions is already well studied and employed in fine-grained action recognition (missing related literature, e.g. FineGym, FineAction, etc.), while comparison and differentiation are lacking in this paper.

4. The core concept of the Multi-modal Action Knowledge mechanism is to introduce a novel video-text alignment strategy. In this alignment framework, the loss function incorporates the cosine similarity between a video clip and its most closely corresponding text segment. A potential concern with this approach is its impact on the advantages derived from the variety of fine-grained textual descriptions, which are crucial for enhancing model generalization. For instance, if multiple video clips repeatedly align with a limited number of high-quality text segments while neglecting a broader range of other texts, the diverse information encapsulated in those overlooked texts might not be effectively integrated into the network's learning process. Consequently, this could lead to an issue of excessive redundancy in the utilization of fine-grained textual descriptions, potentially undermining the model's ability to generalize across different contexts.

5. The proposed method, AP-CLIP, does not outperform baseline methods across all datasets (e.g. Open-VCLIP on HMBD51). Also it is noteworthy that HMDB51 is a relatively more fine-grained action recognition dataset.

6. Ablations and Analyses on ``novel action recognition" is limited.

**Suitability:**

3

---

### Meta-Review · Area_Chair_TjpG · 2024-07-02

**Recommendation:** Accept (Poster)
**Confidence:** 4

**Metareview:**

This paper proposes the Semi-Supervised Camouflaged Object Detection (COD) framework for the first time to address the training challenges caused by noisy manual labels. After discussion, the reviewers noted that the paper has many strengths, including being clearly written, having standardized drawings, and presenting a well-understood motivation and method. Additionally, the pixel-level loss reweighting method is particularly clever. In summary, this paper meets the acceptance standards of the MM conference and is recommended for a poster presentation.